# Variability Management in Self-Adaptive Systems through Deep Learning: A Dynamic Software Product Line Approach

Oscar Aguayo [1] , Samuel Sepúlveda [1,*] and Raúl Mazo [2]

1   Departamento de Ciencias de la Computación e Informática, Centro de Estudios en Ingeniería de Software, Universidad de La Frontera, Temuco 4811230, Chile; oscar.aguayo@ufrontera.cl
2   Pôle Sciences et Technologies de l'Information et de la Communication (STIC), École Nationale Supérieure de Techniques Avancées Bretagne, 29200 Brest, France; raul.mazo@ensta-bretagne.fr
*   Correspondence: samuel.sepulveda@ufrontera.cl

**Abstract:** Self-adaptive systems can autonomously adjust their behavior in response to environmental changes. Nowadays, not only can these systems be engineered individually, but they can also be conceived as members of a family based on the approach of dynamic software product lines. Through systematic mapping, we build on the identified gaps in the variability management of self-adaptive systems; we propose a framework that improves the adaptive capability of self-adaptive systems through feature model generation, variation point generation, the selection of a variation point, and runtime variability management using deep learning and the monitor–analysis–plan–execute–knowledge (MAPE-K) control loop. We compute the permutation of domain features and obtain all the possible variation points that a feature model can possess. After identifying variation points, we obtain an adaptation rule for each variation point of the corresponding product line through a two-stage training of an artificial neural network. To evaluate our proposal, we developed a test case in the context of an air quality-based activity recommender system, in which we generated 11 features and 32 possible variations. The results obtained with the proof of concept show that it is possible to manage identifying new variation points at runtime using deep learning. Future research will employ generating and building variation points using artificial intelligence techniques.

**Keywords:** variability; self-adaptive systems; dynamic software product lines; MAPE-K; deep learning



## 1. Introduction

The software industry has grown enormously and has had to meet the high demands of customers and users regarding the benefits and quality of its products and services [1]. To respond to these new requirements, software engineering (SE) has been formulated as the discipline responsible for all aspects of software production through proposals that address how to develop products and manage the projects and resources involved [2]. Software constantly evolves due to repeated technological changes and the diversification of its needs, either by the client or the execution environment [3]. Software systems have moved away from traditional software systems to self-adaptive ones, which can modify themselves to meet their changing requirements or environment at execution time [4]. Such adaptation extends the software's ability to be subject to change, a concept known in SE as variability [3,5]. One approach to managing the aspects mentioned above is dynamic software product lines (DSPL), which corresponds to a conceptual framework for managing variability in systems that can automatically adapt themselves to changing contexts. One of the most common techniques to implement the adaptation is by reconfiguring the systems' artifacts or attributes when the system or its execution environment changes.

The approach proposed by the DSPL specializes in managing the system variability before and after the running software system [5]. The above allows for generating mixed approaches by incorporating various system states, called variation points, which are related to the environment's static properties before runtime, and others are linked to

the dynamic properties at runtime based on adaptation rules [6]. Therefore, managing variability throughout the life cycle of a DSPL is a central task, where the user, application, or generic middleware can perform these tasks manually or automatically, thus allowing software components to be dynamically added or removed while executing [7].

Variability in self-adaptive systems using the DSPL approach is usually managed adaptation using the MAPE-K control loop [8]. This loop provides a framework for the logical adaptation of a system based on four phases, monitoring, analysis, planning, and execution, which are executed in a given order and access a shared knowledge component, where a record of the events in each phase is maintained [9]. The MAPE-K control loop used to manage a software system presents several problems associated with using this approach, such as the need for significant training data and low initial performance due to online learning of the approach [5]. These problems are similar to those encountered in machine learning techniques and the need for massive data, such as those reported using machine learning in big data for the agricultural area [10].

DSPLs have had an impact in software engineering, particularly in improving the adaptability and efficiency of software systems and the hardware that supports them. For example, in the context of self-adaptive systems, DSPLs enable continuous reconfiguration of software products to adapt to constantly changing environmental contexts, addressing real-time aspects of reconfiguration processes [11]. Furthermore, in the realm of data protection and trusted execution, DSPLs have enabled the adaptation of an application's binary code so that only relevant features are stored in protected memory at any given time, which is crucial in hardware such as Intel's Software Guard Extensions (SGX) [12]. In cyber-physical systems (CPS), DSPLs combine performance models with Multi-Objective Evolutionary Algorithms (MOEAs) to make decisions, improving efficiency through transfer learning, which allows sharing previously acquired knowledge and applying it to similar systems, mastering up to 71% of solutions without transfer learning [13]. Moreover, automata learning, a fundamental technique for building behavioral models, has been extended to evolving systems and variability-intensive systems applicable to the DSPL field, demonstrating its versatility and applicability in various engineering contexts [14]. In the automotive industry, for example, DSPLs have facilitated variability management in the development of parking brake systems, enabling the identification of reuse opportunities and variations early in the product development cycle, leading to less engineering effort and higher quality and more reliable solutions [15]. This approach has been adapted to the specific needs of model-based systems engineering in the automotive industry, implementing suitable means to represent and operate with variability, demonstrating the adaptability and relevance of DSPL in specific industrial contexts [16].

Within the challenges in managing variability using DSPL, there is a need to support evolution regardless of the system domain, such as using various implementation techniques or modeling approaches in DSPL [8]. In addition, DSPLs must ensure system consistency while executing, i.e., that the execution environment has only the features analyzed in the system domain enabled, supporting the visualization of variability in the running system or the model [17]. In the case of self-adaptive systems, variability management in this type of software system is complex because it requires changes at runtime, where such changes may be based on external requirements, either from the execution environment or from its stakeholders [3]. Thus, these changes can become a problem since it is required to manage every possible reconfiguration the system could have [3]. Regarding the challenges related to dynamic variability identified from a systematic mapping of the literature [8,18] proposes to continuously use online learning algorithms to refine the performance influence models to runtime context specifications. In the case of real-time reconfigurations, the challenges lie in defining new software products that meet these requirements at runtime without human intervention [17,19]. Variability management of a DSPL is performed based on variation points corresponding to a set of features selected within the domain to be deployed according to compliance with an adaptation rule [17].

In this article, we address the management of dynamic variability by identifying new variation points at runtime, thus enabling the autonomous management of a self-adaptive system, supporting the problem mentioned above by [3,17,19]. The present work aims to directly address variability management in self-adaptive systems by generating a framework, studying runtime system reconfigurations to identify new points of variation, and extending the range of adaptability of the execution domain. This proposal will allow a self-adaptive system to manage each possible adaptation of the system by binding an adaptation rule using machine learning models. In the context of this work, our research question is as follows: What is the potential of using artificial intelligence techniques to automatically identify new points of variation while the system is at runtime in a DSPL?

The remainder of this paper is structured as follows. We present the background of the DSPLs and their context in Section 2. Section 3 presents some related work on SPLs or DSPLs in variability management. Section 4 presents the methodology used to build and evaluate the FMweb-K framework. Section 5 presents the FMweb-K framework with the stages related to analyzing variation points by deep learning. Sections 6 and 7 present the results and the discussion, respectively. Finally, Section 8 presents the conclusions and future work.

## 2. Background

This section presents the theoretical framework for DSPL variability management. In Section 2.1, we present DSPLs and how they manage variability in software systems. Finally, Section 2.2 presents self-adaptive systems and the MAPE-K control loop.

### 2.1. Dynamic Software Product Lines

DSPLs are a software production approach based on software product lines (SPLs) that share a common architecture and reusable components [5]. They can automatically adapt and configure themselves in response to changes in the operating environment, system requirements, or user preferences, all while the system operates. The primary purpose of DSPLs is to increase the flexibility and responsiveness of software systems. By enabling runtime adaptation, these systems can handle unforeseen conditions, take advantage of new opportunities, and mitigate emerging threats without the need for manual intervention; here, one of their main objectives is to maintain or improve system functionality and performance in the face of change while reducing the costs and time associated with deploying traditional upgrades and configurations [17]. The main innovation offered by DSPL is runtime adaptability: while traditional systems may require downtime to apply changes or upgrades, DSPL-based systems can adapt quickly, which is crucial for critical systems that require high availability [8]. This approach facilitates continuous customization and system optimization, as changes can be made based on real-time user or system behavior rather than relying solely on predefined configurations. In addition, DSPL engineering reduces the need for long-term predictions and allows systems to evolve organically in response to usage patterns and environmental changes [7].

DSPLs comprise two main stages: the engineering cycle and runtime variability management [7]. The engineering cycle aims to identify common characteristics and define a robust software architecture. This process establishes essential variability mechanisms for future configurations and reconfigurations, emphasizing the design of modular and reusable components, culminating with the development and testing of these components, ensuring their ability to dynamically modify the behavior or structure of the system through elements such as plugins or microservices. Runtime variability management focuses on monitoring the system and its operating environment, covering aspects ranging from resource utilization to critical external factors. This comprehensive management allows the system or a human operator, equipped with accurate data, to proactively determine the need for reconfigurations, guided by rules, policies, or machine learning algorithms. In the event of critical changes, the system has the ability to automatically reconfigure itself

at runtime, adjusting functions, resources, or operational parameters, enabling real-time adaptability without interruptions in service or functionality.

The configuration of a particular product in a product line is the process of selecting a valid and complete set of variants to link to variation points [20]. In the case of a DSPL, the variation points are used to describe all possible states of the system, where a set of conditions must be met. These conditions are called adaptation rules, which define how to change the state of the software between different variations [17]. Figure 1 presents the change in system variability when an adaptation rule is activated, and the variation point is modified according to some change in environmental requirements.

Over the years, various methodologies have been developed to address system variability management using the DSPL approach. These methodologies can be classified into three main scenarios [17]. The first scenario focuses on customizing system changes manually; for example, by manually updating the software code base. The second scenario allows changes to be made manually while the customization process is automated, as in DevOps processes. The third scenario provides an autonomous execution environment that allows automation of variability changes and subsequent reconfigurations, thus providing an optimal environment for self-adaptive systems.

Due to the difficulty of anticipating all the variability required by self-adaptive systems, we can consider the use of DSPLs to facilitate self-adaptation in this type of system. Several approaches express changes in variability as a function of the context [21] (e.g., in feature modeling [22]). Other approaches consider machine-learning-based reconfiguration management using the MAPE-K control loop, which is an architectural approach to system variability management [23].

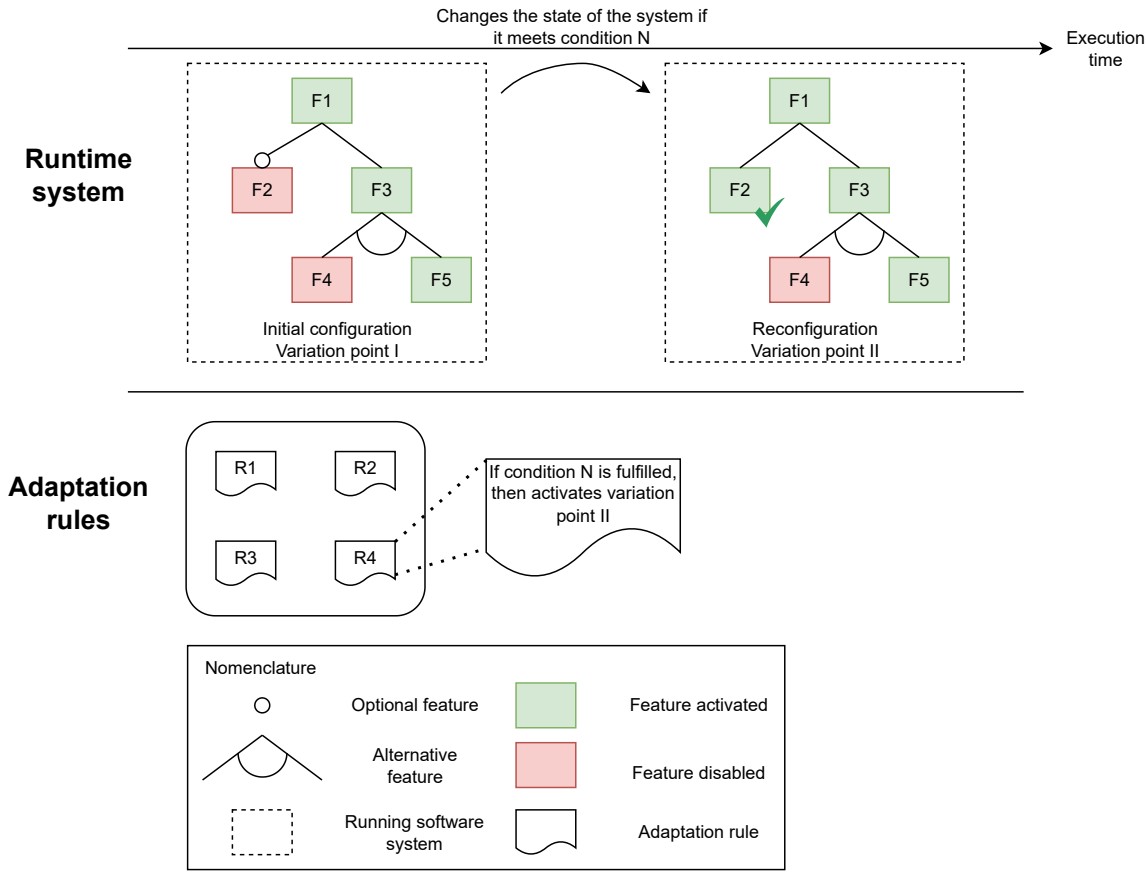

**Figure 1.** Design and application of variation points in DSPL. Adapted from [17].

## 2.2. Self-Adaptive Systems

Self-adaptive systems correspond to a closed-loop system with feedback intended to adjust to runtime changes autonomously without third-party intervention [24]. In addition to the "feedback loop", which allows the system to evaluate the effectiveness of its adaptations and make adjustments to optimize future responses, self-adaptive software systems must meet two fundamental principles [3]. On the one hand, the external principle stresses that such a system must autonomously manage changes and uncertainties caused by the demands of the environment, the system's internal conditions, and its predefined objectives, implying a proactive adaptation to variations without external intervention. On the other hand, the internal principle specifies that a self-adaptive system comprises two essential elements that manage its interaction with the environment and its adaptability. These principles include components dedicated to constantly monitoring the system and its context; i.e., sensors and a control logic that decides how to respond based on this information. In addition, it has actuators that execute necessary changes in real-time, adjusting operability according to the circumstances detected. These components and mechanisms ensure that the system maintains efficient, resilient, and autonomous functionality, even in dynamic and uncertain environments.

The MAPE-K control loop is widely used to manage architectures for self-adaptive systems with DSPLs [8]. MAPE-K is a control loop that manages runtime variability through a four-stage iterative cycle and a shared knowledge node [9]. This information flows to the analysis stage, where the system's current state is evaluated, determining whether corrective actions are required. If intervention is required, the planning stage comes into play, developing a strategic action plan that dictates the measures necessary to achieve the desired state. This plan materializes in the execution stage, where actions are implemented and monitored to ensure their correct application. The knowledge repository serves as a connection between all the modules, acting as a central storage and intelligence hub that is accessible to all parts of the loop. This repository contains the collected and processed data, as well as essential information such as architecture models, policies, and change plans. This information is essential for making informed decisions and proactive adaptations.

## 3. Related Work

This paper proposes FMweb-K, a variability management-oriented framework using the MAPE-K control loop and deep learning that can be compared with other variability management proposals in DSPL. It is possible to manage a system's variability through a product line using the following tools.

DyMMer 2.0 is a tool that allows modeling and generating variation points in DSPL [25]. Its main features are the creation and edition of feature model, and the MAPE-K control loop is the primary adaptation mechanism. It has a web tool developed in Vue.JS. Additionally, it is proposed to occupy machine learning models that aim to classify the maintainability of the [26] feature model.

Moskitt4SPL is a software application that facilitates the creation of both static and dynamic product lines. It enables users to model their product lines in a straightforward manner [27]. Its main features include feature model editing, configuration model generation, and DSPL context modeling. It is built on the Eclipse platform and is open source.

VariaMos is a web-based tool that utilizes microservices to enable the specification, multi-language modeling, and multi-solver reasoning of SPL and DSPL projects [28]. In addition, it enables users to create their own engineering languages in a straightforward manner. The instances of these languages are then used to represent the domain engineering assets in a way that they can be analyzed, verified, simulated, configured, and federated to create static and dynamic software product lines. From an interoperability point of view, VariaMos allows exporting XLS configuration files and JSON configuration files, importing JSON configuration files, and saving/loading models from XML files.

S.P.L.O.T. [29] is a web application that allows the creation of feature models and offers some reasoning functionalities on the models. It uses a binary decision diagram engine

and a Boolean satisfiability problem solver (SAT solver) to perform various analyses such as counting the number of possible variation points, calculating the degree of variability of a model, checking the consistency of a model, and detecting common features and dead features (features without accessibility). In addition, the tool offers an extensive repository of feature models, where several previously created models can be found.

Feature IDE [30] is a tool that, in addition to allowing the creation of feature models, it allows users to configure the features and then convert these configurations into products of the related SPL, which makes it possible to create a package of a software system from a chosen set of features [31].

In contrast, our FMweb-K framework not only allows the creation of feature models but also allows the definition and use of dynamic contexts to manage the variability of self-adaptive systems at runtime. In addition, our framework allows (i) managing self-adaptive systems that implement a MAPE-K architecture, and (ii) identifying, through the use of neural networks, variation points that were not contemplated during the design of the product lines.

Finally, the proposals that partially manage run-time variability are those that allow modeling and simulating the behavior of the system in case of a reconfiguration. Unlike the other proposals, FMweb-K aimed at managing self-adaptive systems, so it is not feasible to design the variability visually as in other proposals. Table 1 summarizes the comparison between the various proposals (✓ managed DSPL stage, ∼ partially managed).

**Table 1.** Comparison of proposals to manage variability in DSPL stages.

| Proposal | Engineering Cycle | Runtime Variability Management |
|---|---|---|
| DyMMer 2.0 | ✓ | ∼ |
| FeatureIDE | ✓ | ∼ |
| Moskitt4SPL | ✓ | |
| S.P.L.O.T | ✓ | ∼ |
| VariaMos | ✓ | ∼ |
| FMweb-K | ∼ | ✓ |

## 4. Methodology

To develop the FMweb-K proposal, we used an adaptation of the Design Science methodology, proposed by Wieringa [32], which aims to address research in software engineering and information systems. The methodology focuses on the design and construction of artifacts and the evaluation of their usefulness and effectiveness in solving problems within a specific domain, which is ideal for this field of study, as it focuses on the creation and evaluation of artifacts designed to fulfill a specific purpose, developing practical and tangible solutions [33]. The Design Science methodology promotes an iterative approach, continuously improving artifacts based on feedback and test results. Iterative and agile development is fundamental in software engineering, corresponding to a common practice to adapt to rapid changes in requirements and technologies [34]. The methodology has four fundamental stages: (i) problem investigation or implementation evaluation, (ii) treatment design, (iii) treatment validation, and (iv) treatment implementation.

### 4.1. Problem Investigation

This stage aims to investigate a problem that can be improved before we design a technological artifact, in which the requirements to develop such an artifact still need to be identified. Therefore, our objective at this stage was to identify the current modeling languages and techniques of DSPL engineering, and their limitations in managing variability. We pursued our objectives by following the guidelines of Petersen et al. [35] in order to identifythe primary approaches, architectures, and challenges in variability management for DSPLs.

*4.2. Treatment Design*

This stage is to design the artifacts that provide a solution to the problems encountered in the initial phase, reducing the gap between what is achieved and what is desired. In this context, the specific objectives associated with this stage correspond to determining the procedure, tools, and assets necessary to develop FMweb-K. We designed the proposal in the context of the reference architectures presented by [5,7,17].

*4.3. Treatment Validation*

This stage aims to assess whether the design of the solution will bring stakeholders closer to the defined goals, where this assessment is only a prediction and will be verified once the solution is implemented. For model validation, we opt for the Delphi method using expert criteria [36] and proof of concept methodology [37].

In software engineering and artificial intelligence, the Delphi methodology has been used in several proposals to validate a proposal, such as to validate the design of the effort estimation in agile software projects through deep learning [38]. Furthermore, approaches based on machine learning and software product lines have been presented to detect whether an information visualization can be potentially confusing and misinterpreted, which was validated by adapting the Delphi methodology and a proof of concept [39].

In this context, the objectives associated with this stage correspond to the validation of the design of the framework for the management of feature modeling, the creation of variation points, and adaptation rules for the management of variability in DSPLs. Additionally, we wish to validate that the MAPE-K control loop allows the management of changes in variability from the selection and deployment of variation points, as well as the implementation of a deep learning model to identify new variation points with their respective adaptation rule.

A group of experts in software engineering and software product lines was involved in the application of the Delphi method. These experts were essential to validate whether our proposed architecture, once implemented, would effectively address the central challenge of our study [40]. We use the methodology to validate our design focused on variability management of self-adaptive systems using DSPLs, from the selection and deployment of variation points to the implementation of a deep learning model to identify new variation points with their respective adaptation rules. Through two rounds of questionnaires and discussions, we explored and evaluated various architectures in artificial intelligence, including different algorithms for supervised learning, semi-supervised learning, and more specialized approaches such as convolutional neural network adaptations or artificial neural networks with transfer learning. We used the proof of concept methodology to validate with experts whose approach supported us in solving the problem associated with a low initial amount of data [37]. This iterative and collaborative process allowed us to refine our proposals, evaluating the feasibility of our proposal with artificial neural networks and learning transfer.

*4.4. Treatment Implementation*

This stage constitutes the implementation of the design previously validated through expert criteria. In this context, the objectives associated with this stage correspond to the implementation of the different components for variability management in the DSPL, such as feature modeling, variation points, the MAPE-K control loop and the artificial neural network to identify new variation points using an iterative-incremental development methodology[41]. Additionally, we performed a new iteration of the design science cycle to evaluate our implementation. In this context, the objective associated with this stage corresponds to the realization of a set of tests to evaluate the correct functionality of the proposal, both in managing variability and in analyzing new points of variation through the proof of concept methodology [37].

## 5. FMweb-K Framework

The origins of the proposal go back to the FMxx feature modeling tool, where we allowed modeling variability in a product line and linked it to software artifacts for subsequent deployment [42]. We base the proposal on two stages that can be visualized in Figure 2. The first stage is in charge of managing runtime variability of a DSPL by means of feature models, the MAPE-K control loop, variation points, and adaptation rules. The second stage seeks to analyze new points of variation in runtime, through a deep learning model, based on two stages and transfer of learning between these two stages.

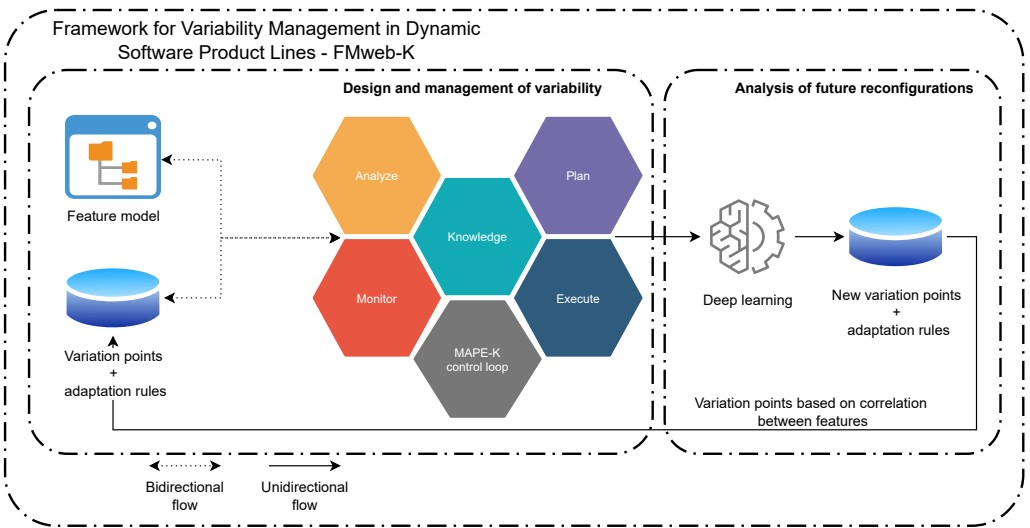

**Figure 2.** FMweb-K framework.

The systematic mapping study that we conducted aimed to capture approaches, methodologies, or design patterns for variability management in DSPLs and self-adaptive systems. This mapping included the analysis of the types of approaches used to manage runtime variability, and to maintain system stability and reliability during the constant evolution of variability. In addition, it was possible to identify the principal errors, difficulties, or challenges in (i) developing DSPLs and (ii) adapting the systems derived from these DSPLs [8].

Next, Section 5.1 presents the steps of the proposed framework for managing variability by identifying new variation points. Section 5.2 presents the test case and its setup. Finally, Section 5.3 presents the preliminary validation in which we train a deep learning model in the context of the test case to identify new variation points.

### 5.1. Variability Management through MAPE-K Loop and Deep Learning

Based on identifying needs through the development of the systematic literature mapping, we identified the main modules that the new proposal must guarantee, among which we highlight the implementation of the MAPE-K control loop to manage system reconfigurations and a deep learning module to identify variation points at runtime. The proposal includes six functionalities, which are detailed below. The details of these capabilities are presented in Figure 3.

- Feature model generation: generation of feature models with their respective dependencies and constraints.
- Variation points generation: generation of configuration models from the previously defined feature model.
- Variation point selection: selecting a variation point for subsequent automatic deployment.
- Software product deployment: deployment of the variation point configuring the execution environment with the selected features.

- System monitoring via MAPE-K loop: monitoring of the software system, looking for possible reconfiguration requests and resolving them at runtime.
- Variation point analysis by deep learning: identify a variation point by incorporating an artificial intelligence component, allowing us to obtain new system states in addition to those initially defined in the variation point generation functionality.

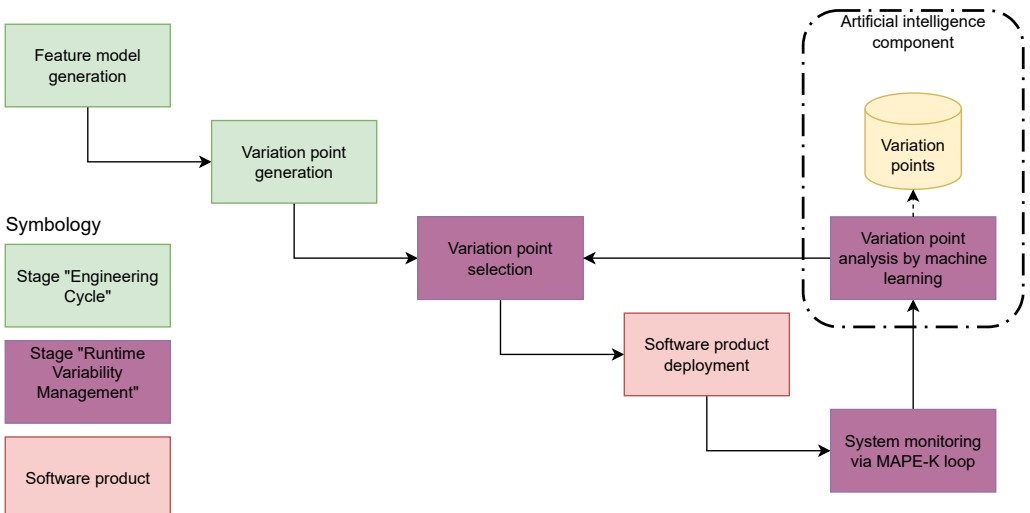

**Figure 3.** FMweb-K functionalities.

The MAPE-K control loop enables the system to be monitored, while deep learning is used to analyze variation points. However, the data input to train the deep learning model is limited to the initial set of variation points, which is usually small and thus not suitable for machine learning tasks. Thus, the solution to this limitation is to employ a transfer learning approach. Transfer learning allows the use of the knowledge acquired by a previously trained model on a large and diverse dataset [43]. In the context of DSPL, few variation points are defined at the beginning, making it difficult to identify new states with machine learning techniques. In such cases, transfer learning approaches are often used; for example, one study focused on Multi-Objective Evolutionary Algorithms (MOEAs) in DSPL and proposed transfer learning to improve the efficiency and quality in the generation of configurations [13]. Transfer learning offers the flexibility to adapt models to new tasks relatively quickly. It is beneficial in rapidly evolving fields where models must continually adapt to new data types or problems. In deep learning, this approach uses previously acquired knowledge from one domain or problem to apply it to a different but related domain or problem [44]. In this context, it allows for improved efficiency and effectiveness in training new models by leveraging existing knowledge rather than starting from scratch. This deep learning technique contains two stages: pre-training and tuning, which are detailed below [44].

- Pre-training: In this stage, a machine learning model is trained on a large and diverse dataset, acquiring general valuable knowledge in various tasks. This pre-trained model can be an image classification model a natural language processing model, among others.
- Tuning: We adapt the pre-trained model to a specific problem or domain of interest in this stage. Adaptation between machine learning models usually involves adjusting some layers of the deep learning model, replacing them with new and more appropriate ones, and then training the model with a problem-specific dataset. The tuning process allows the model to specialize and improve its performance on the exciting task.

Through the permutation of domain features, we obtain all the possible variation points that a feature model can possess and then predict the various adaptation rules linked to each variation point through the artificial neural network. Finally, the system

must be able to receive any adaptation rule and link it to a variation point. The linkage of an adaptation rule to a variation point will be through a function that minimizes the absolute difference between a target value of the dependent variable, which in this case are the variation points, and the values of the independent variable, which correspond to the adaptation rules previously linked to each variation point. Figure 4 presents the activities associated with the use of artificial intelligence in our proposal.

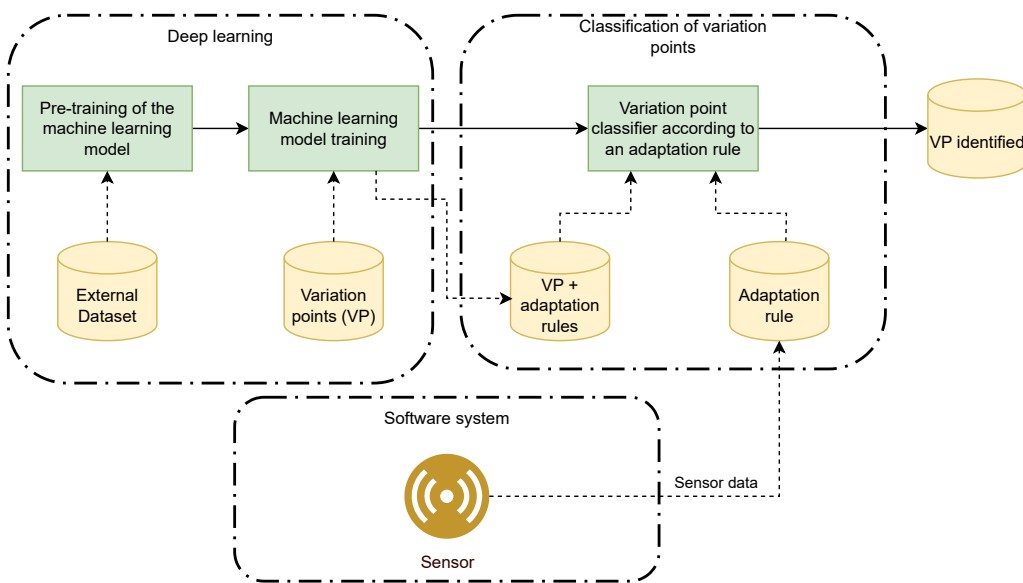

**Figure 4.** Variation point prediction phases.

To realize changes in system variability at runtime, we occupy the MAPE-K loop to manage software systems under a microservices architecture, precisely a set of Docker containers, where each container corresponds to a particular feature of the problem domain. The MAPE-K control loop allows activating the features of a variation point and deactivating those that do not correspond to the current state of the system from the Docker library available in Python.

### 5.2. Test Case Scenario

To test our proposal, we performed a test case in the context of an air quality-based activity recommender system, in which, based on a feature model, we generated variation points and their subsequent variability management. Figure 5 presents the feature model for the activity recommender system, which has the following properties; here, the features that are always active in the domain are presented with a check mark, and features that appear with a question mark indicate that they must be configured at the point of variation.

*Air quality viewer* is a mandatory feature in the model and *Firewood use restriction viewer* is an optional one. The feature *Tourism* is mandatory in the model and has *Closed environments* or *Open environments* as a mandatory feature. In the case of the feature *Closed Environments*, it requires *Firewood to use restriction viewer* for its operation. Likewise, the feature *Open environments* require the feature *Sports* for its operation. The feature *Sports* is optional in the model. The *Entertainment* feature is an optional feature of the model and optionally has the *Family Entertainment*, *Senior Entertainment*, and *Adult Entertainment* features.

The architecture of the test case is based on microservices, specifically on micro frontends, since it is an architectural approach focused on managing services in a decentralized way. From identifying new points of variation with deep learning and variability management with the MAPE-K loop, we encapsulate each feature towards a specific container, which allows us to activate and deactivate components easily.

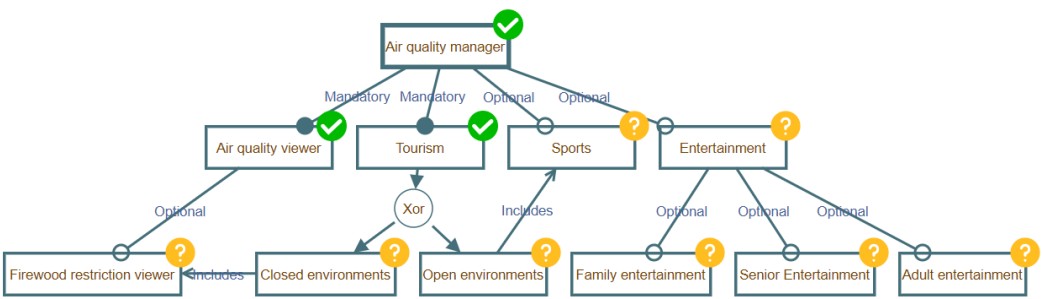

**Figure 5.** Feature model for test case.

*5.3. Preliminary Validation*

From the test case, we will preliminarily validate the proposal, using two-stage training to identify variation points using two datasets and deep neural networks. To carry out this process, we used a dataset of 223,508 records associated with atmospheric emissions from land transport in Chile during the year 2020. A first data filtering was performed according to the context of the test case so that both datasets are in a similar and more limited context, including only records related to the problem domain, obtaining a total of 84,079 records to train the artificial neural network. Subsequently, we applied data scaling to the dataset using normalization, which consists of pre-processing data to adjust the feature scales to be in a typical range, usually [0, 1] or [−1, 1] [45]. In this case, an absolute maximum value scaler (MaxAbsScaler), which transforms the sparse data by dividing by the maximum absolute value in each feature [46].

We built the first deep neural network model for system pre-training. The network has several dense layers with different sizes and activation functions to perform model training. When training the neural network, it obtains the weights of the first hidden layer of the model and creates a feature extractor from this layer. This feature extractor is a separate model that takes as input the data in the same format as the original neural network and produces as output the activations of the neurons in the first hidden layer. The extractor allows to take advantage of the knowledge acquired by the neural network during its training, where the central idea is that the first hidden layer has learned to recognize certain essential features of the input data. By using only this layer as a feature extractor, new input data can be transformed into a feature set containing relevant information learned by the neural network. Figure 6 presents the training phases for the deep neural network. The neural network architecture that we occupy for this preliminary validation is based on an adaptation of a convolutional neural network in terms of layers and activation function since it allows us to identify images from patterns, which is a similar context to what we wish to implement in the identification of variation points [47]. In addition, in the first instance, we occupy the Relu activation function due to its regular use in software engineering [48], starting with a network of 128 and 64 neurons, ensuring a lower training error [49].

Next, we build a new model that transforms the variation point data to have the same dimension as the air emission set data, where this model is used to transform the training and test sets of the variation point set. The dataset associated with the variation points are normalized to the range [0, 1] [46]. For each feature, the minimum value is subtracted and then divided by the range (maximum–minimum), looking for them to be in a specific, finite range, which in this case is to describe whether a variation point feature is on or off.

Finally, we build a deep neural network model using the extracted features for the small dataset. The network has several dense layers with different sizes and activation functions to train the model with the training data from the set of variation points. After training the neural network, we perform a regression-type prediction for all the variation points obtained from the permutation of features and their relationships. Figure 7 presents the training phases of deep learning training in the transfer of learning.

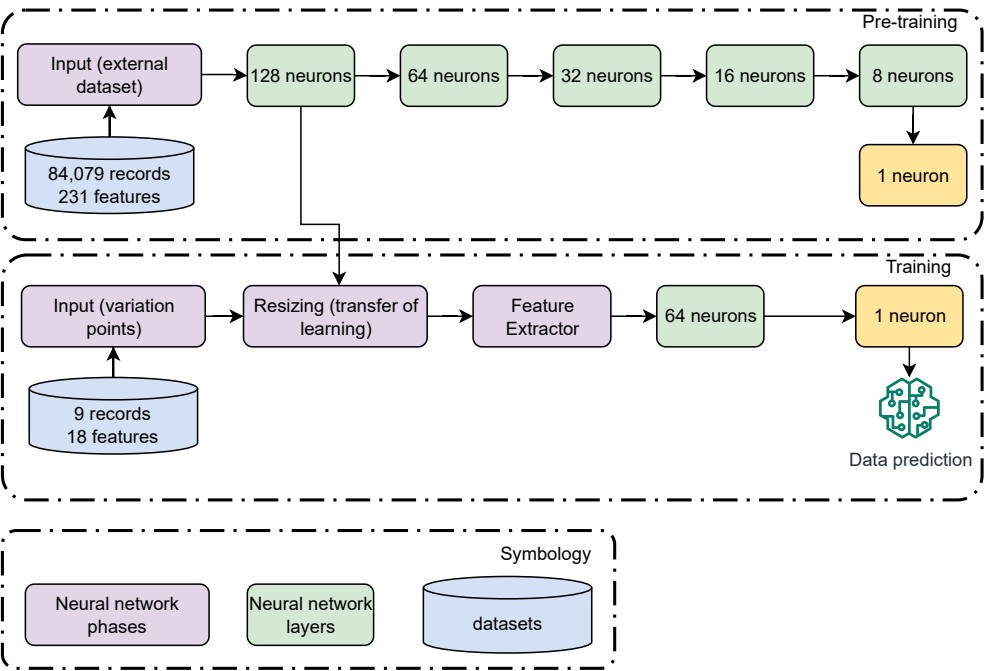

**Figure 6.** Phases for the artificial neural network.

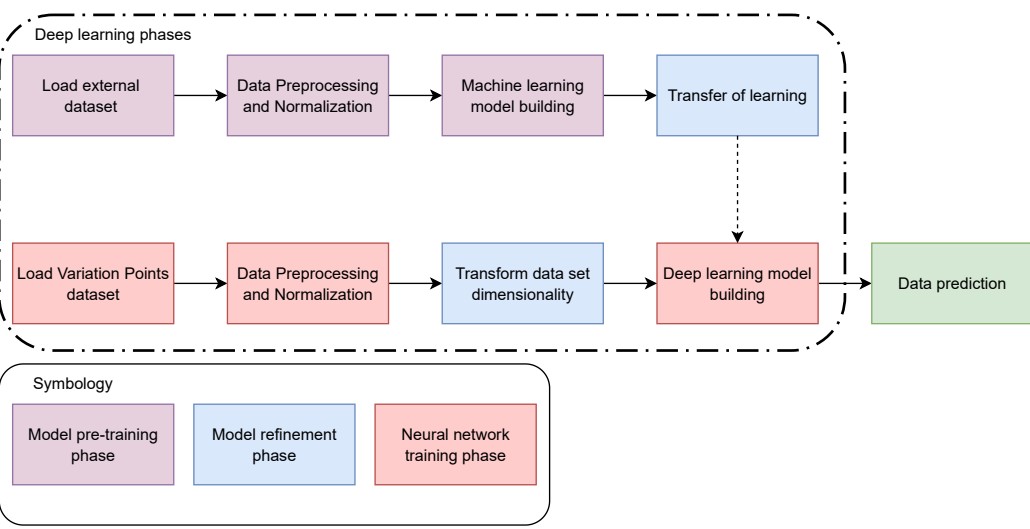

**Figure 7.** Phases for deep learning.

## 6. Results

Through developing the proposal framework, we built a series of functionalities to enable the variability management of a self-adaptive system using DSPL. The main results on generating variation points are visualized in Section 6.1. Section 6.2 presents the results of identifying variation points using deep learning. Section 6.3 presents the results on variability management in DSPLs using the MAPE-K control loop and Docker. Finally, Section 6.4 presents a proof of concept.

### 6.1. Generation of Variation Points

We built the feature model in Python, using a graph to record each feature and its relationships. Relationships of type *OR, XOR, Mandatory, and Optional* indicated the cycle and hierarchy of each network node as parent and child features. Dependencies of type *XOR, Requires, and Excludes* indicate the constraints that must be applied by the features when the system creates a variation point.

We managed the generation of variation points by permutation of features, obtaining all the variation points corresponding to the model and then subtracting the constraint type relations to obtain the total possible configurations of a model. Each variation point created was stored in a CSV file to describe all possible variation points of a problem domain.

### 6.2. Identification of Variation Points through Deep Learning

To identify variation points using deep learning, we use a set of 24 possible configurations in the neural network, seeking to obtain the best performance during the pre-training and training of the model. To choose the configuration of the deep neural network to be implemented, we analyzed the test cases presented in Table 2.

In the context of the artificial neural network implementation, we use different regularization techniques to avoid overfitting, such as L1 , L2, and the batch normalization function [50]. L1 regularization, also known as Lasso regularization, adds a penalty equal to the absolute value of the coefficients of the network weights. This penalization can lead to some weights becoming zero, which is helpful for feature selection in large models [50]. The L1 regularization is added to the neural network cost function. The equation for L1 regularization is:

$$L1 = \lambda \sum_i |w_i|, \tag{1}$$

where $\lambda$ is a regularization parameter and $|w_i|$ are the weights of the network. The sum of the absolute values of the weights implies that some weights may become zero, which helps in feature selection [51].

On the other hand, L2 regularization, known as Ridge regularization, adds a penalty equal to the square of the magnitude of the coefficients [52]. This penalization tends to distribute the error among all the weights, often resulting in a smoother and more generalizable model. Similar to L1, L2 is added to the cost function. The equation for the L2 regularization is:

$$L2 = \lambda \sum_i w_i^2, \tag{2}$$

In the L2 function, $\lambda$ is also a regularization parameter and $w_i$ are the weights. The sum of squares of the weights distributes the error across all weights, which can lead to a more generalizable model [52].

Batch normalization is a different technique used to improve artificial neural networks' speed, performance, and stability. It is applied to the input of each layer and normalizes the data to have a mean of zero and a variance of one [53].

$$\hat{x} = \frac{x - \mu_B}{\sqrt{\sigma_B^2 + \epsilon}}, \tag{3}$$

where $\mu_B$ is the batch mean, $\sigma_B^2$ is the batch variance, and $\epsilon$ is a small number to avoid division by zero [54]. After this normalization, two learnable parameters, $\gamma$ (scale) and $\beta$ (offset) [55], are applied for each feature:

$$y = \gamma \hat{x} + \beta, \tag{4}$$

Finally, these parameters are adjusted during training and allow the batch normalization to maintain the representative capacity of the network [55].

The tests to be performed are based on the evaluation of the model, comparing the error and accumulated error in each training iteration. Each instance to be tested will have a five-layer pre-training model. The first layer has 128 neurons; the second layer has 64 neurons; the third layer has 32 neurons; the fourth layer has 16 neurons; and the fifth layer has 8 neurons. This layer configuration seeks to reduce the dimensionality of the data; that is, it seeks to reduce the number of random variables under consideration by

obtaining a set of primary variables and simplifying the data without losing too much information [56].

**Table 2.** Neural network performance test cases.

| Optimizer | Training Rate | Regulatory Function |
|---|---|---|
| Adagrad | 0.01 | Batch Normalization (BT) |
| Adagrad | 0.01 | L1 |
| Adagrad | 0.01 | L2 |
| Adagrad | 0.001 | Batch Normalization (BT) |
| Adagrad | 0.001 | L1 |
| Adagrad | 0.001 | L2 |
| Adam | 0.01 | Batch Normalization (BT) |
| Adam | 0.01 | L1 |
| Adam | 0.01 | L2 |
| Adam | 0.001 | Batch Normalization (BT) |
| Adam | 0.001 | L1 |
| Adam | 0.001 | L2 |
| RMSprop | 0.01 | Batch Normalization (BT) |
| RMSprop | 0.01 | L1 |
| RMSprop | 0.01 | L2 |
| RMSprop | 0.001 | Batch Normalization (BT) |
| RMSprop | 0.001 | L1 |
| RMSprop | 0.001 | L2 |
| SGD | 0.01 | Batch Normalization (BT) |
| SGD | 0.01 | L1 |
| SGD | 0.01 | L2 |
| SGD | 0.001 | Batch Normalization (BT) |
| SGD | 0.001 | L1 |
| SGD | 0.001 | L2 |

Regarding the regulatory functions, which will be used only in the pre-training model, a range of 0.001 will be used for L1 and L2. In the second model, to train the neural network from the learning transfer, it will have an additional layer of 64 neurons. In the case of optimizers, Adagrad, Adam, and RMSprop are optimizers that maintain adaptive learning rates. In contrast, the SGD optimizer is a linear optimizer that is more concerned with optimization since it updates the model parameters at each iteration through only one network sample.

In the evaluation of the 24 possible cases, we sought to obtain the best performance in the models, where in the case of the Adagrad optimizer, the pre-training that had the best performance in terms of error and cumulative error was with training rate of 0.01 and the regularizing function batch normalization. Figure 8 presents the error and cumulative error of the Adagrad optimizer in the pre-training of the model.

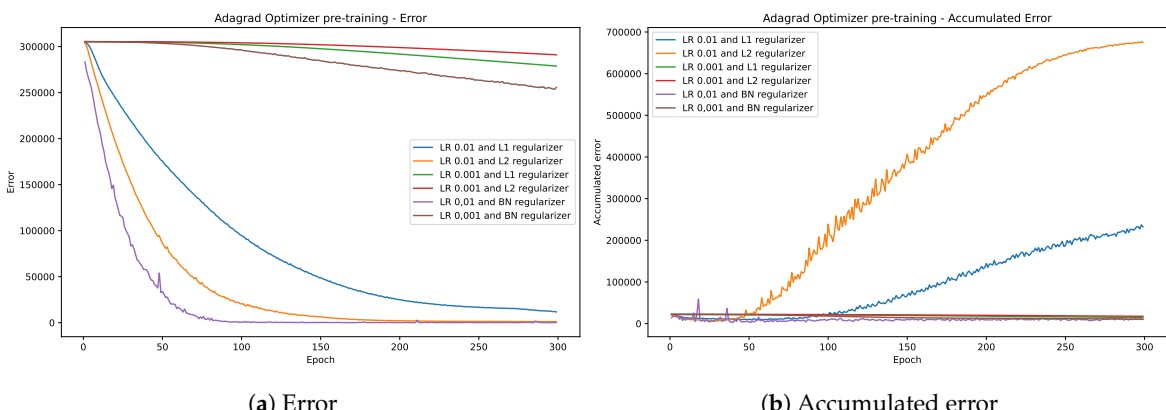

(**a**) Error          (**b**) Accumulated error

**Figure 8.** Pre-training performance with Adagrad Optimizer.

The Adagrad optimizer that achieved the best results after the learning transfer was the same as the configuration displayed above, with a learning range of 0.01 and the batch Normalization regularizing function. Figure 9 presents the error and the accumulated error of the Adagrad optimizer in training the model after learning transfer in pre-training.

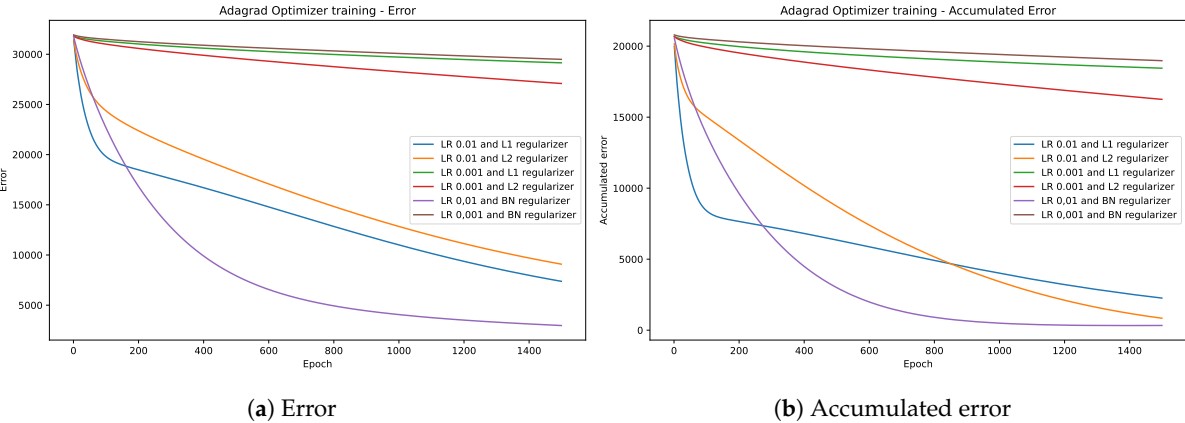

(**a**) Error           (**b**) Accumulated error

**Figure 9.** Training performance with Adagrad Optimizer.

In the case of the Adam optimizer, the pre-training that performed best in terms of error and cumulative error was with training ratios of 0.001 and the regularizing function L2. After learning transfer, the best-performing Adam optimizer training was the same as the configuration shown above, with a learning range of 0.001 and regularizing function L2. The pre-training of the RMSprop optimizer that performed best in error and cumulative error was with a training rate of 0.001 and regularizing function L1. The RMSprop optimizer training performed best after transfer learning with the learning range 0.01 and the batch Normalization regularizing function. However, it performs similarly for models with the learning range 0.001 and the batch normalization regularizing function. Finally, in the case of the SGD optimizer, no pre-training model performed well, all having over 300,000 errors. The training of the machine learning model using the SGD optimizer after the learning transfer evolves improvable in the data prediction; however, the great majority remains with an error close to 9000. The SGD optimizer with the learning range of 0.01 and the L2 regulator presented an error elevation during the first iterations, starting with an error of up to 500,000,000. However, after a few iterations, it remains equally at an error close to 9000. Appendix A contains the plots of the error-based performance of the neural network training with the Adam, RMSprop, and SGD optimizers.

According to the performance analysis based on error and cumulative error, we choose the configuration associated with the Adagrad optimizer and the batch normalization regulator function because of its pre-training and post-transfer learning model training performance.

The identification of variation points using artificial neural networks is divided into two stages, present in the following git repository (accessed on 31 October 2023). These two stages consisted of training the network, identifying the points of variation, and linking these points of variation to the given adaptation rules.

The first stage consists of the collection of all the variation points contained in a specific problem domain; we need to statically define some initially defined variation points to allow the model to be trained. This dataset is limited and requires a learning transfer from another machine-learning model to train the model correctly. For this purpose, a first neural network was developed with a dataset similar to the problem domain of the feature model to identify patterns in feature selection for machine learning. After the learning transfer, we train a second neural network with the dataset belonging to the defined variation points with their respective adaptation rule. Finally, after 24 experiments with different configurations in the neural network, we used the Adagrad optimizer, with a learning range of 0.01 and a batch normalization regulator function. After training the automatic

learning model, we identify the adaptation rule associated with each record of the set of variation points obtained from the feature permutation.

We obtain a variation point from an adaptation rule in the second stage. To accomplish this goal, we use a function that uses the absolute difference as a metric to evaluate which feature configuration most closely resembles the target value of the independent variable to be evaluated. In our case, the set of variation points identified in the deep machine learning process. Finally, the configuration that minimizes this absolute difference is selected.

### 6.3. Variability Management in DSPL Using the MAPE-K Control Loop

We define the variability management from the MAPE-K control loop, which we developed using Python and the Docker API. In this context, we simulate the state of a sensor to establish the value of an adaptation rule and to identify a point of variation that satisfies that adaptation rule, which allows us to activate or deactivate features of the problem domain. Each feature corresponds to a different Docker container for the test case presented in Section 5.3, allowing efficient integration and management through a micro frontends architecture.

In the test case, we also display active (marked in green) and deactivated (marked in red) features. Figure 10 presents the feature model of the test case with its respective feature status.

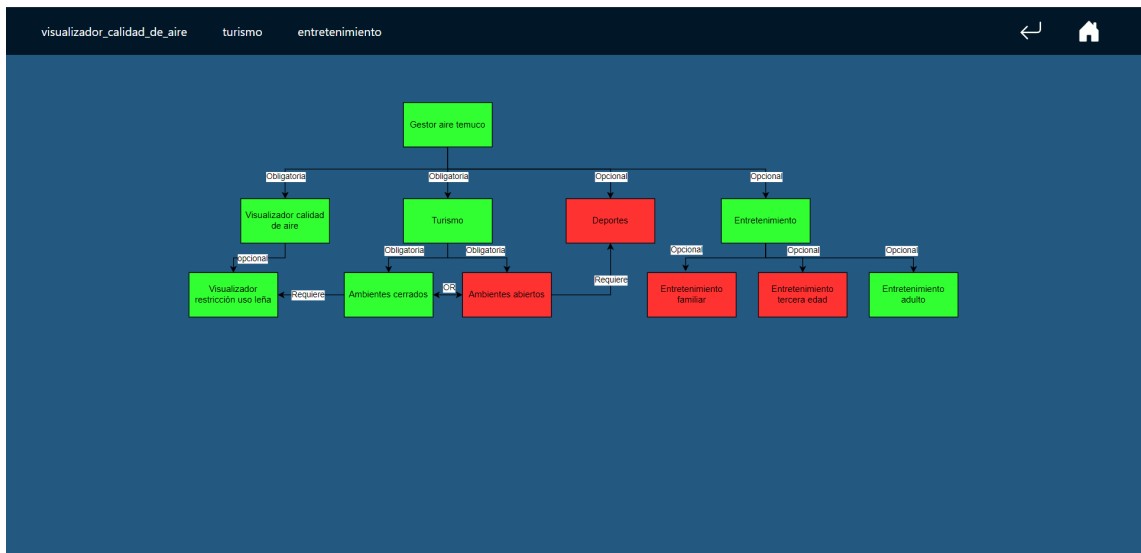

**Figure 10.** Display of activated and deactivated features.

### 6.4. Proof of Concept

In the implementation evaluation stage of the design science methodology, we performed a series of validations to corroborate the correct performance of the implementation of the proposal. We present the results of the proposal's implementation in a test case oriented to a recreational activities recommender system based on air quality with its results associated with the proof of concept applied to the implementation of the project.

The test case development is based on a monolithic micro frontend architecture since the server raises and accesses the resources available for each microservice, so only the server renders the resources. The main components we develop are in the Host app, which is the main component that manages the other micro frontends and the global configurations of the prototype, such as the security policies of the web content, the imported libraries, and the paths assigned to the components. In addition to the above, it contains the SVG format diagram that represents the prototype's structure at the sub-application level and the development of its activation states in runtime. The Host app component manages the components through import maps, where we set the URL of the components, assigning them a port on which each service runs. Additionally, we added

a navbar, which corresponds to the dynamic navigation bar of the prototype, located in the upper section. This bar contains the links to the active sub-applications when the user interacts with the interface. These links and the SVG diagram are updated periodically without the user needing to update it directly from their browser. The component also has two buttons: a "Back to Home" button that redirects the user to the main screen and a "Back to Back" button that redirects the user to the last accessed sub-application. For a more intuitive design, we use SVG icons for these buttons. Finally, we managed the overall design of the prototype in another container called Styleguide.

Figure 11 presents the architecture defined for the test case, containing four Docker containers that manage the test case requests and deployment. In addition, it has ten Docker containers that represent the various features of the test case, which are available in a Git (accessed on 20 November 2023) repository. The architecture consists of the API contained in a docker as the rest of the structure, which has the power to modify every 2 min of service available on the network. The front end receives the information from the API. It dynamically updates the navigation bar every 200 milliseconds, making visible only the services available at the time of consultation so we ensure that the client can only view them.

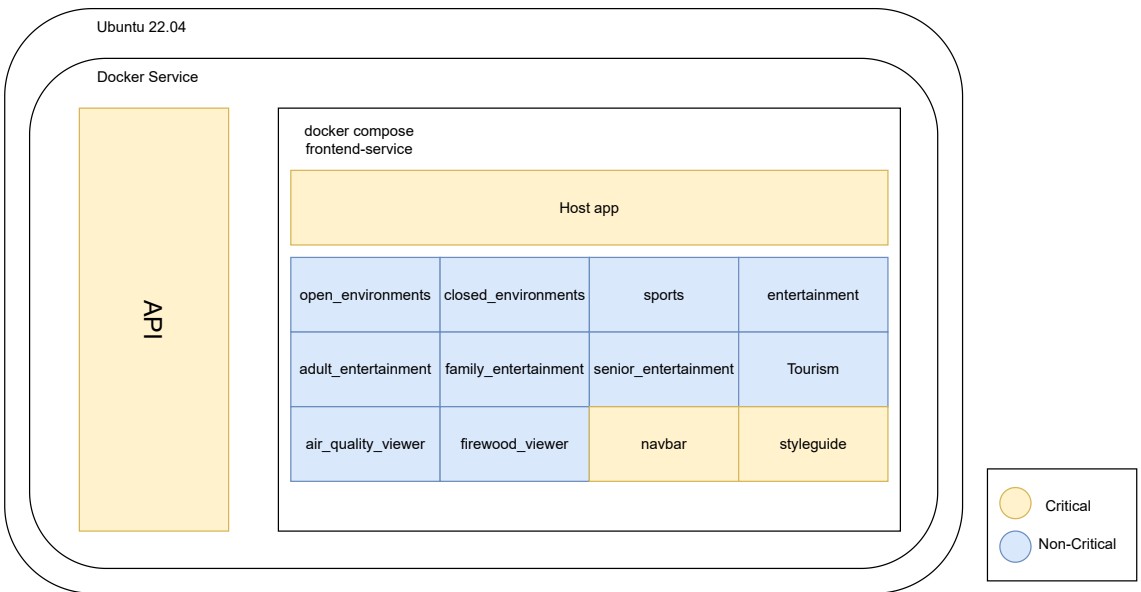

**Figure 11.** Test case architecture.

We conducted a proof of concept to preliminarily validate the proposal, divided into two segments, as presented in Tables 3 and 4. The first segment contemplates the analysis of variation points through machine learning techniques, explicitly using deep neural networks to evaluate prediction performance. The second segment is related to the generation of software reconfigurations. It refers to the implementation of the MAPE-K loop from the sensor simulation, the search for variation points related to the adaptation rule mapped to the sensor, and its subsequent execution of the reconfiguration.

According to the results obtained from the tests, we can answer the research question formulated at the beginning of the paper by the following statement that it is possible to automatically identify new runtime variation points in a DSPL. With the incorporation of machine learning models and the MAPE-K control loop, it is possible to manage a self-adaptive system using a DSPL approach.

**Table 3.** Decision table for validating variation point analysis through deep learning.

| N° | Case Description | Input Section | | Output Section |
|---|---|---|---|---|
| | | Input Variables | State before Testing | Expected Results |
| 1 | Feature permutation to obtain all associated variation points | Feature model | Feature modeling with its respective relationships | Set of associated variation points |
| 2 | Pre-training of machine learning model | Pre-training dataset | Not applicable | Trained machine learning model |
| 3 | Learning transfer between machine learning models | Pre-trained machine learning model | Dimensionality transformation of variation point set | Machine learning model with learning transfer |
| 4 | Training of machine learning model | Initial dataset of variation points with its adaptation rule | Machine learning model with learning transfer | Total set of variation points with its associated adaptation rule |
| 5 | Variation point prediction | Adaptation rule | Random forest model trained with total set of variation points with the associated adaptation rule | Identified variation point |

**Table 4.** Decision table for validating the generation of software reconfigurations.

| N° | Case Description | Input Section | | Output Section |
|---|---|---|---|---|
| | | Input Variables | State before Testing | Expected Results |
| 1 | Searching for variation point according to sensor value | Adaptation rule value | Machine learning model to the software system | Selection of the variation point linked to the adaptation rule |
| 2 | Identification of variation point in execution | Selected variation point | Variation points linked to the software system | Identification of active variation point in the software system |
| 3 | Execution of variation point | Active and chosen variation points | Variation points linked to the software system | Activation of chosen variation point and deactivation of active variation point |

## 7. Discussion

The use of deep learning has been increasingly seen in the area of software engineering, as well as in the domain of software product lines, implementing various proposals, such as the implementation of defect prediction techniques to improve the quality assurance activities of computer systems or the search for improvement in the use of machine learning through reuse techniques in the implementation of artificial neural networks [57,58]. Likewise, product line engineering has also contributed to the electronics area, making it possible to reuse the building blocks of the emerging behaviors of artificial neural networks to train robotic controllers more efficiently [59]. In this context, we based the development of the framework FMweb-K on the use of deep learning to identify variation points with the adaptation rule and the MAPE-K control loop to manage the variability of the system. Before applying deep training to identify the variation points with their respective adaptation rules, we permute all the features in the model to obtain the total set of possible states. We then store them in a CSV file to have the input resource to identify the trigger of that variation point by deep learning. We designed a two-stage architecture from that initial approach, mixing two machine learning techniques called a hybrid machine learning system [60]. In validating the architecture design using the Delphi and proof-of-concept methodologies, we performed several tests on various architectures based on artificial intel-

ligence. The experts associated with the Delphi methodology do not share direct conflicts of interest in this research. We tested several techniques to identify variation points from the proof of concept methodology.

The first stage consists of identifying the total set of variation points. By incorporating transfer learning and a collection of variation points with their corresponding adaptation rules, it is possible to recognize adaptation rules for potential variation points in the corresponding domain. The second stage involves linking an adaptation rule with a variation point containing features and their states. Through a machine learning model for multi-class classification of results, we seek to obtain multiple feature states from a given adaptation rule. In our proposal, initially, the classification algorithm receives the dataset predicted by the artificial neural network as input. Then, from the adaptation rule given to the system, the model must deliver the associated variation point. Since the initial dataset was minimal, with 32 samples in the test case context (see Section 5.3 for more details), we opted for a relation that occupies the absolute difference between each adaptation rule to classify the closest variation point.

Figure 12 presents the performance of the classification model according to the F1 metrics and the area under the multi-class ROC curve (AUC-ROC) [46]. The F1 score is a metric that combines model accuracy and sensitivity into a single value, taking values between 0 and 1, where 1 indicates perfect model performance, and 0 indicates poor performance. As for the multi-class AUC-ROC, this metric measures the ability of a multi-class classification model to correctly distinguish between all classes, which is in the range of 0 to 1, where a value of 1 indicates perfect model performance in multi-class classification and a value of 0.5 indicates performance similar to a random choice. We evaluated five models: random forests, decision trees, K-nearest neighbors, and MLPClassifier, a variant of an artificial neural network called a multilayer perceptron [61]. In summary, the performance of each model was similar to theirs, denoting the lack of a more considerable amount of input data to improve classification, as the evaluations indicate a performance similar to a random choice [62,63].

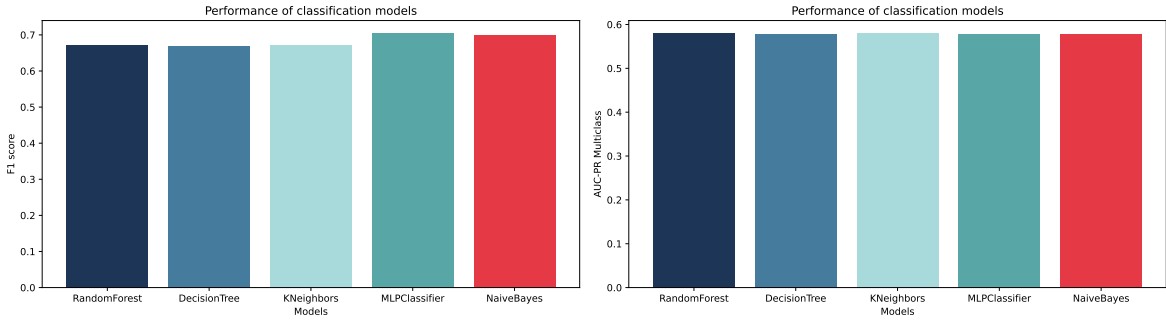

(**a**) F1 performance assessment        (**b**) ROC curve assessment

**Figure 12.** Performance evaluation for supervised machine learning in variation point classification.

When using the stochastic gradient descent (SGD) optimizer to create an artificial neural network, it was found to be less effective than the Adagrad, Adam, and RMSprop optimizers. This is because SGD does not have adaptation techniques or the ability to automatically adjust the learning range, which the other optimizers possess [64]. As a result, the initial impact of random initialization of the weights may be more pronounced in SGD training, which may lead to a more variable initial loss behavior, as we show in Figure A6. The first iterations resulted in an error of 500,000, leading to a training anomaly due to the random initialization of the weights in the neural network connections, which could have an initial effect on the loss behavior [64]. Figure 13 presents a new training of the neural network with the atypical case of the SGD optimizer and the L2 regularizer function. With this change, we saw a slight improvement in the early stages of the training process, which eventually led to the same error rate and total error as those discussed in Section 6.2.

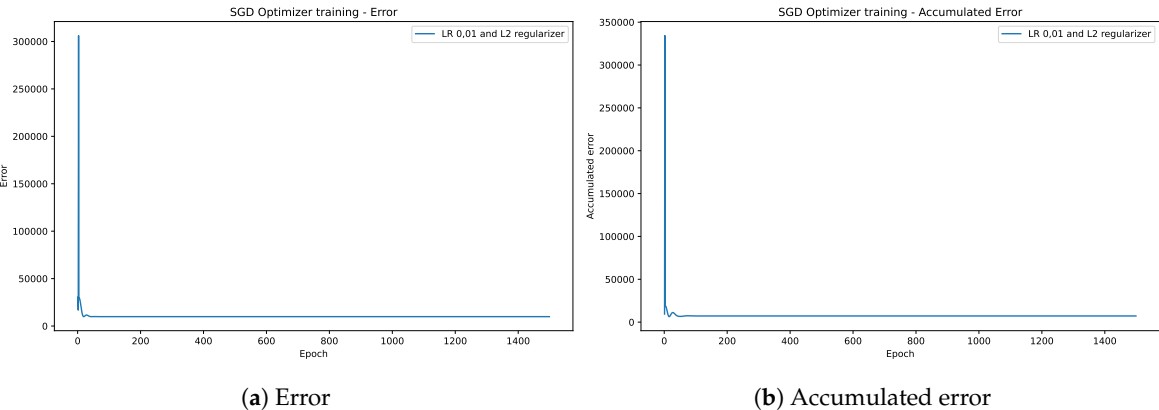

(**a**) Error　　　　　　　　　　　　　　　　　　(**b**) Accumulated error

**Figure 13.** Neural network performance evaluation with SGD optimizer and L2 regularizer.

In evaluating model performance, the choice of the Adagrad optimizer in combination with the batch normalization controller function proved to be decisive in achieving lower uncertainty in terms of error and cumulative error. This configuration effectively handled the adaptive learning rate, resulting in more stable and consistent convergence than other optimizers such as Adam, RMSprop, and SGD. The latter presented challenges in converging to a stable minimum, probably due to the complexity of the parameter space involved in learning transfer. During neural network pre-training, high error was initially observed. However, the model significantly improved over epochs, culminating in low and cumulative errors. This evolution in performance suggests that the network effectively matched the data, learning essential features without overfitting. The implementation of batch normalization allowed the normalization of each layer's inputs, thus reducing the problem of internal covariate change, improving the training process, and contributing to a higher overall stability of the model. In addition, the applied learning transfer allowed the model to adapt better and generalize from prior knowledge, which is evident in the decrease in error throughout training. This strategy ensured that the model learned from data specific to the current problem and leveraged patterns and features learned from previous tasks, resulting in more robust learning and reduced uncertainty.

We developed the test case based on a micro-frontend architecture, which provides several benefits to web frontend development, including support for different technologies, autonomous cross-functional teams, and independent development, deployment, and management [65]. In this context, developing a test case in this architecture benefits the development of a self-adaptive system since it fulfills the main characteristics of DSPL engineering: be able to add or remove features at runtime, and be able to modify the variability set of the system without the need to redeploy the entire system. Platforms such as Apache Karaf provide the capability of dynamic variability management due to their use of dynamic software containers that can be associated with a base project [66]. However, this implementation was discarded because of the complexity of programming the modules and working with Karaf. In addition to the micro frontends, we used Docker containers to manage each test case feature in a different container, which allows us to segment and isolate each service as proposed by [67]. We implemented the container manager directly through the Docker API for Python, which allows us to incorporate the deep learning model, the MAPE-K control loop, and component management in the same software component. In the case of other machine learning approaches, evidence suggests that a reinforcement machine learning system can be employed to manage autonomous vehicle applications with the help of Docker, Kubeadm, and the machine learning model in RSUs [68].

The results obtained in this research project have certain limitations. Even if the proposal allows for identifying new variation points at runtime, it is limited to a previous definition of variation points with their respective adaptation rules since it is required to have an initial dataset to train the machine learning model and distinguish valid recon-

figurations. With little data to train the machine learning model, additional techniques must be used to make predictions, such as transfer, semi-supervised, or reinforcement learning techniques. On the other hand, it is required to obtain an additional dataset that is of a similar context to the test case to be managed in order to pre-train the deep machine learning model and perform the learning transfer. To not require a dataset similar to the problem domain, a technique change is required, such as an adaptation to reinforcement learning algorithms: an agent learns to make decisions by acting in an environment and receiving feedback through rewards or punishments.

The evaluation of the neural network training is limited to occupying the activation function Relu and the optimizers Adagrad, Adam, RMSprop, and SGD with learning ranges of 0.01 and 0.001, respectively. We propose to extend the range of learning and optimizers in future work by occupying subvariants of the above optimizers such as momentum, Nesterov accelerated gradient (NAG), Adadelta, Adamax, and Nadam [64].

## 8. Conclusions

This paper introduces FMweb-K, a system for managing variability in DSPLs. It enables the identification of new points of variation at runtime from a shared set of potential system states. Additionally, it allows the software system to be reconfigured in accordance with adaptation rules, providing a dynamic context for self-adaptive systems using the MAPE-K control loop and deep learning.

While developing our proposal, we adapted the design science methodology proposed by Wieringa [32]. Although we knew that variability management in DSPLs for self-adaptive systems was an important and promising topic, we also knew that this topic had great challenges. To explore the advances and challenges of this topic, we performed a systematic mapping of the literature from 2010 to 2021. This resulted in the selection of 84 articles related to the problem domain, which revealed various methodologies, architectures, and challenges in variability management for self-adaptive systems using DSPLs.

In the solution design, we propose a system based on six stages: feature model generation, variation points, selection of a variation point, and runtime variability management through deep learning and MAPE-K control loop. We design a feature permutation to obtain all the variation points of a problem domain and obtain an adaptation rule for each variation point of the system through a two-stage training of an artificial neural network. Finally, in the solution evaluation, we develop a test case in the context of an air quality-based activity recommender system, creating a feature model with 11 features and 32 possible variation points. We evaluated the artificial neural network with four optimizers with two learning ranges. In addition, three regularization techniques, L1, L2, and batch normalization, were employed, resulting in 24 experiments to determine the most successful configuration that included the Adagrad optimizer and the batch normalization regularization technique. Finally, we performed a proof of concept to validate the proposal, where eight functional tests were presented to the proposal, completing the tests presented.

In future work, we propose to incorporate a meta-model in the feature model to extend the range of possibilities in the context of self-adaptive systems, with several adaptation rules for the same variation point, as well as to extend the range of evaluation of the presented deep machine learning. In addition, we encourage extending the range of study of identifying new variation points, analyzing various machine learning techniques, and increasing the range of optimizers, regulators, and learning ranges in the case of artificial neural networks. Finally, artificial intelligence techniques can support several areas of the product line, such as building and deploying new software products automatically, identifying variant performance, supporting the MAPE-K control loop in configuration tasks, and driving the generation of variants of customer interest through natural language processing.

**Author Contributions:** Conceptualization, O.A. and S.S.; methodology, S.S. and O.A.; software, O.A.; validation, O.A., S.S. and R.M.; formal analysis, O.A. and S.S.; investigation, O.A.; resources, O.A. and S.S.; data curation, O.A.; writing—original draft preparation, O.A.; writing—review and editing, S.S. and R.M.; visualization, O.A.; supervision, S.S. and R.M.; project administration, S.S.; funding acquisition, S.S. and O.A. All authors have read and agreed to the published version of the manuscript.

**Funding:** Oscar Aguayo thanks to Universidad de La Frontera, Vicerrectoría de Investigación y Postgrado, research project DIUFRO DI24-0112. Samuel Sepúlveda thanks to ANID—Fondecyt de Iniciación, research project Nº 11240702.

**Institutional Review Board Statement:** Not applicable.

**Informed Consent Statement:** Not applicable.

**Data Availability Statement:** Data are contained within the article.

**Conflicts of Interest:** The authors declare no conflicts of interest.

## Appendix A

Figures A1 and A2 present the training results of the artificial neural network in the pre-training and training of the neural network with the Adam optimizer.

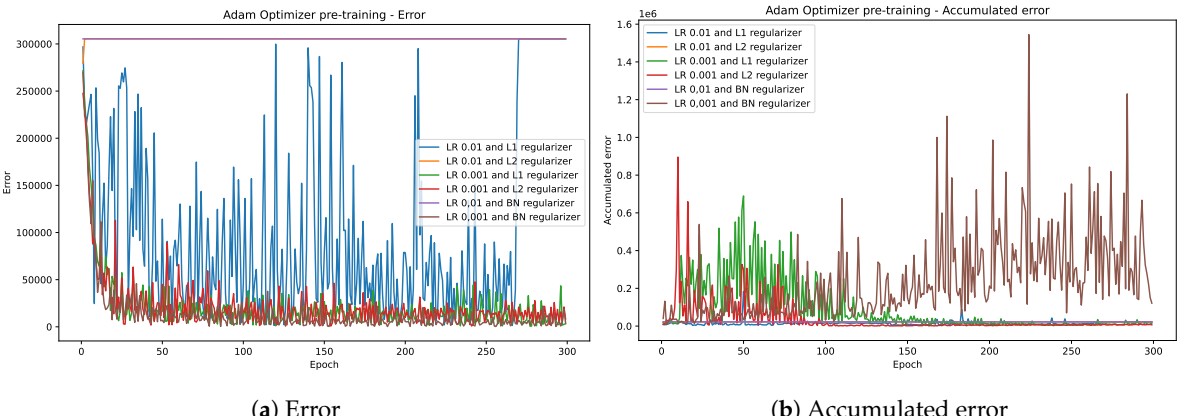

(**a**) Error                                                                                   (**b**) Accumulated error

**Figure A1.** Pre-training performance with Adam Optimizer.

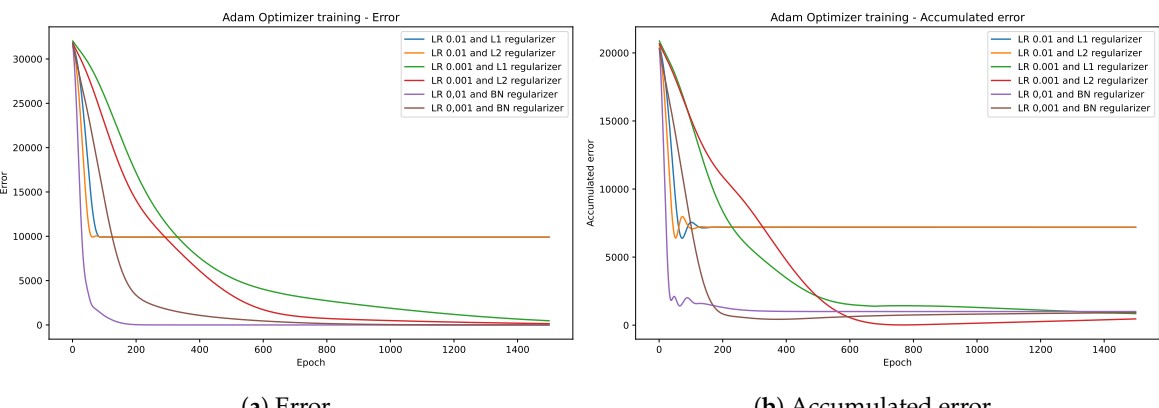

(**a**) Error                                                                                   (**b**) Accumulated error

**Figure A2.** Training performance with Adam Optimizer.

Figures A3 and A4 present the training results of the artificial neural network in the pre-training and training of the neural network with the RMSprop optimizer.

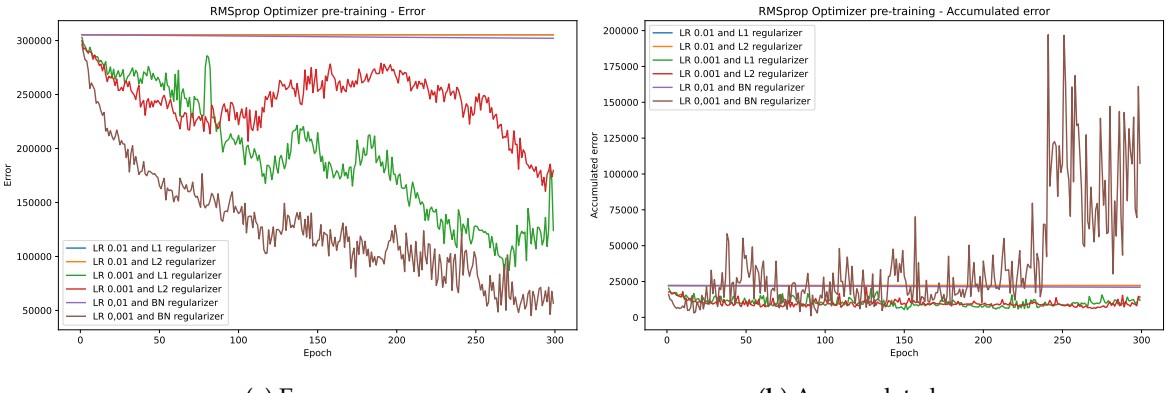

(**a**) Error

(**b**) Accumulated error

**Figure A3.** Pre-training performance with RMSprop Optimizer.

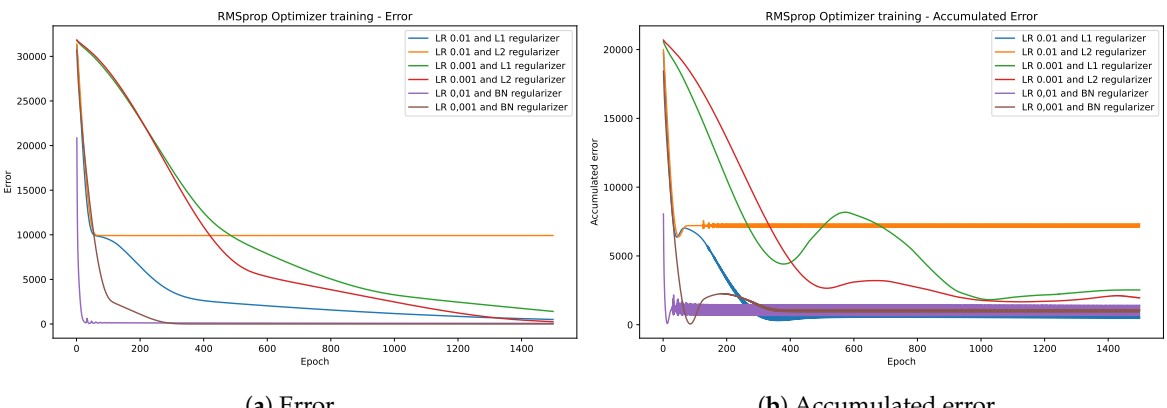

(**a**) Error

(**b**) Accumulated error

**Figure A4.** Training performance with RMSprop Optimizer.

Finally, Figures A5 and A6 present the training results of the artificial neural network in the pre-training and training of the neural network with the SGD optimizer.

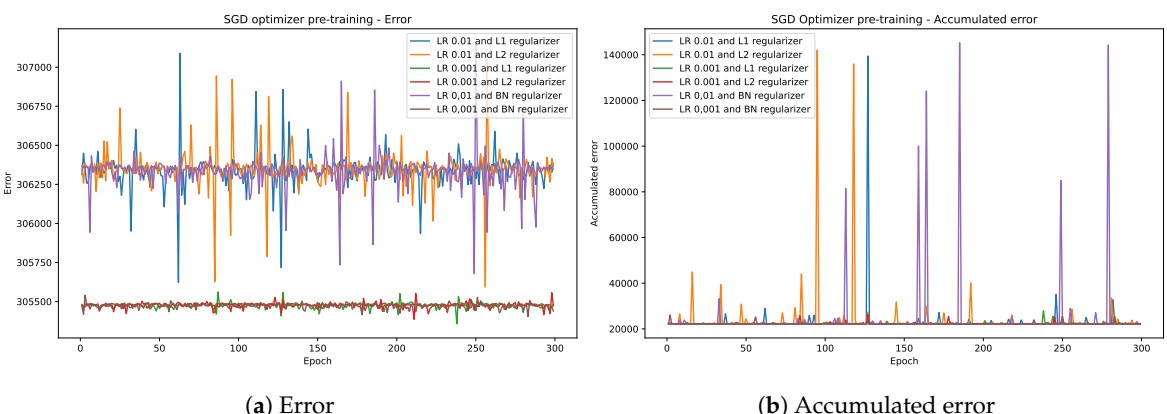

(**a**) Error

(**b**) Accumulated error

**Figure A5.** Pre-training performance with SGD Optimizer.

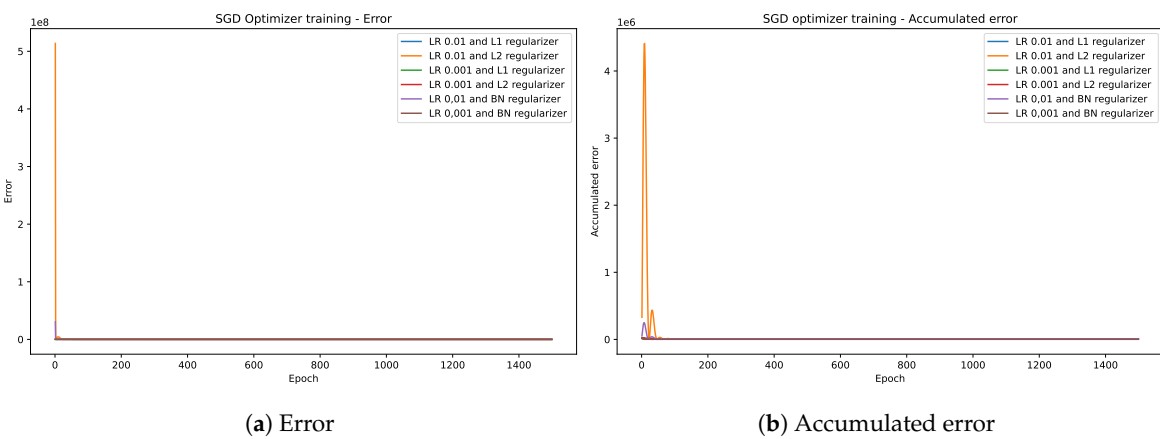

(**a**) Error            (**b**) Accumulated error

**Figure A6.** Training performance with SGD Optimizer.

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
