# Peer review of "Variability Management in Self-Adaptive Systems through Deep Learning: A Dynamic Software Product Line Approach"

_electronics, doi:10.3390/electronics13050905_

Round 1

Reviewer 1 Report

Comments and Suggestions for Authors

1. The description of the methodology lacks some key details. For instance, the study mentions the use of the Delphi method for treatment validation but does not elaborate on how this method was applied, who the experts were, or how consensus was reached.

2. The study introduces the Design Science methodology for artifact development. However, it does not explicitly justify why this methodology was chosen over others.

3. The treatment validation stage relies on the Delphi method and expert criteria. It's crucial to discuss the limitations and potential biases associated with expert opinions. Additionally, considering other validation methods, such as empirical studies or user testing, could provide a more comprehensive validation of the proposed framework.

4. The study uses transfer learning in the context of deep learning but does not delve into the specifics of why this approach is chosen or how it contributes to the goals of the study.

5. The study uses a dataset associated with atmospheric emissions from land transport in Chile for training the artificial neural network. However, the rationale for choosing this specific dataset for a variability management problem in DSPLs needs clarification.

6. The test case scenario is specific to an air quality-based activity recommender system. The external validity of the proposed framework in addressing variability management across different domains or applications should be added.

7. Deep learning is well-known and has been used in previous studies i.e., PMID: 36166351, PMID: 34730875. Therefore, the authors are suggested to refer to more works in this description to attract a broader readership.

8. The study should provide sufficient details about the implementation of the proposed framework, including code, configurations, and parameters used.

9. The study lacks a discussion of potential limitations and challenges associated with the chosen approach.

10. Uncertainties of models should be reported.

11. Quality of figures should be improved.

Comments on the Quality of English Language

English writing and presentation style should be improved.

Author Response

This letter presents the authors’ replies to the reviewers’ comments. First of all, we want to thank the reviewers for their suggestions which have been very useful for improving our manuscript. In this document, we indicate the changes carried out in the paper according to the reviews and suggestions. We would also like to thank the editor for handling the paper.

Should you require any further information or clarification, please do not hesitate to contact us.

For the sake of clarity, we have structured our text so that our modifications appear after each reviewer’s comments. For each modification, we have provided details on the particular section and paragraph of the article that the modification refers to.

Thank you very much again for your careful revision process and the insightful comments of the journal reviewers, which have greatly contributed to improving our manuscript.

Sincerely, 

Oscar Aguayo, Samuel Sepúlveda and Raúl Mazo.

Reviewer 2 Report

Comments and Suggestions for Authors

1) This submission appears to deviate from the intended scope of "electronics." The paper titled "Variability Management in Self-Adaptive Systems through Deep Learning: A Dynamic Software Product Line Approach" would be better suited for publication in a journal focused on "applied science" or another journal dedicated to artificial intelligence.

2) Kindly incorporate motivations from the perspective of "electronics" to strengthen the relevance of your work.

3) Ensure that the complete name of MAPE-K is provided in its entirety during its initial mention in the abstract.

4) Adjust the capitalization of "Dynamic Software Product Lines" to lowercase in line 30.

5) In Figure 1, consider incorporating relationships with "electronics" to enhance its relevance to the intended scope.

6) On line 176, could you provide clarification on the nature and purpose of FMweb-K?

7) Is it appropriate to introduce L1 and L2 on page 12 without providing any accompanying explanation?

8) Are there any mathematical equations included? If not, please include the necessary equations to formulate the problem, present solutions, and evaluate the results.

Comments on the Quality of English Language

1) This submission appears to deviate from the intended scope of "electronics." The paper titled "Variability Management in Self-Adaptive Systems through Deep Learning: A Dynamic Software Product Line Approach" would be better suited for publication in a journal focused on "applied science" or another journal dedicated to artificial intelligence.

2) Kindly incorporate motivations from the perspective of "electronics" to strengthen the relevance of your work.

3) Ensure that the complete name of MAPE-K is provided in its entirety during its initial mention in the abstract.

4) Adjust the capitalization of "Dynamic Software Product Lines" to lowercase in line 30.

5) In Figure 1, consider incorporating relationships with "electronics" to enhance its relevance to the intended scope.

6) On line 176, could you provide clarification on the nature and purpose of FMweb-K?

7) Is it appropriate to introduce L1 and L2 on page 12 without providing any accompanying explanation?

8) Are there any mathematical equations included? If not, please include the necessary equations to formulate the problem, present solutions, and evaluate the results.

Author Response

(The authors gave the same response as above.)

Round 2

Reviewer 2 Report

Comments and Suggestions for Authors

Thank you for the thoughtful revisions. The manuscript is now deemed acceptable for publication.

Comments on the Quality of English Language

Some sentences could be rephrased for improved clarity and conciseness. Ensure that the submission adheres to grammatical standards for a more polished presentation.

Author Response

This letter presents the authors' responses to the reviewers' comments. First of all, we would like to thank the reviewers for their suggestions, which have been useful in improving our manuscript throughout the review process. We would also like to thank the editor for processing the paper.

For any further information or clarification, please do not hesitate to contact us.

Thank you again for your careful review process and for the comments of the journal reviewers, which have contributed greatly to improving our manuscript.
